# Integrated Analysis and Identification of Critical RNA-Binding Proteins in Bladder Cancer

**DOI:** 10.3390/cancers14153739

**Published:** 2022-07-31

**Authors:** Lijiang Gu, Yuhang Chen, Xing Li, Yibo Mei, Jinlai Zhou, Jianbin Ma, Mengzhao Zhang, Tao Hou, Dalin He, Jin Zeng

**Affiliations:** 1Department of Urology, The First Affiliated Hospital of Xi’an Jiaotong University, Xi’an 710061, China; gulijiang1125@outlook.com (L.G.); yhchen@stu.xjtu.edu.cn (Y.C.); xingli_813@163.com (X.L.); myb961217@stu.xjtu.edu.cn (Y.M.); zhoujl@stu.xjtu.edu.cn (J.Z.); 15279191473@163.com (J.M.); zhangmz10@163.com (M.Z.); houtao1994@126.com (T.H.); 2Oncology Research Lab, Key Laboratory of Environment and Genes Related to Diseases, Ministry of Education, Xi’an 710061, China

**Keywords:** bladder cancer, RNA-binding protein, survival, prognosis, tumor microenvironment

## Abstract

**Simple Summary:**

The role of RNA-binding proteins (RBPs) in bladder cancer (BC) remains unclear. Therefore, we analyzed the clinical information and RNA sequencing data from patients with BC and identified RBPs that may be promising predictors of BC.

**Abstract:**

RBPs in the development and progression of BC remains unclear. Here, we elucidated the role of RBPs in predicting the survival of patients with BC. Clinical information and RNA sequencing data of the training and validation cohorts were downloaded from the Cancer Genome Atlas and Gene Expression Omnibus databases, respectively. Survival-related differentially expressed RBPs were identified using Cox regression analyses. A total of 113 upregulated and 54 downregulated RBPs were observed, with six showing prognostic values (AHNAK, MAP1B, LAMA2, P4HB, FASN, and GSDMB). In both the GSE32548 and GSE31684 datasets, patients with low-risk scores in survival-related six RBPs-based prognostic model showed longer overall survival than those with high-risk scores. AHNAK, MAP1B, P4HB, and FASN expression were significantly upregulated in both BC tissues and cell lines. BC tissues from high-risk group showed higher proportions of naive CD4+ T cells, M0 and M2 macrophages, and neutrophils and lower proportions of plasma cells, CD8+ T cells, and T-cell follicular helper compared to low-risk group. AHNAK knockdown significantly inhibited the proliferation, invasion, and migration of BC cells in vitro and inhibited the growth of subcutaneous tumors in vivo. We thus developed and functionally validated a novel six RBPs-based prognostic model for BC.

## 1. Introduction

BC is among the ten leading causes of cancer-related deaths worldwide [1] and account for 3% of new cancer cases worldwide in 2020 [2]. The main therapeutic approaches for BC include chemotherapy and surgery; even with radical surgery, the five-year overall survival (OS) rate is 70%. Cystoscopy, computed tomography, and urine cytology remain the dominant diagnostic and prognostic screening methods for BC; however, these methods are invasive, expensive, and low-sensitive. Recently, researchers have been trying to identify new prognostic markers for BC, which may improve the early diagnosis, progression monitoring, risk prediction, and therapy optimization of BC.

RNA-binding proteins (RBPs) form an essential class of cellular proteins that recognize specific RNA-binding structural domains and interact with RNA and are thereby involved in post-transcriptional regulation, including RNA sequence editing, transport, translation control, shearing, and intracellular localization. More than 1500 RBPs have been found in human genome [3], and increasing evidence suggests that RBPs play major roles in human diseases [4,5]. Mutations and alterations in RBPs have been observed in many tumor tissues and are involved in RNA coding to form ribonucleoprotein complexes [6,7,8,9,10]. RBPs interact with microRNAs [11] and circular RNAs to affect tumor progression and contribute to tumor initiation and growth [12]. Through bioinformatics analysis, Okholm et al. [13] identified nine RBPs that are closely related to the oncogenesis, progression, and metastasis of BC. Wu et al. [14] identified six RBPs via weighted correlation network analysis (WGCNA) and found that HSPG2 acted as an antiangiogenic peptide that inhibited endothelial cell migration and collagen-induced endothelial tube morphogenesis. Furthermore, 12 prognostic RBP signatures have been explored, including CTIF, CTU1, DARS2, ENOX1, IGF2BP2, LIN28A, MTG1, NOVA1, PPARGC1B, RBMS3, TDRD1, and ZNF106. Studies found that MTG1, CTU1, and PPARGC1B gene expressions were related to optimistic prognoses in patients with BC, whereas high RBMS3, DARS2, ENOX1, IGF2BP2, ZNF106, CTIF, and NOVA1 gene expressions were related to pessimistic prognoses [15]. However, the exact and functional role of RBPs in the development and prognosis of BC remain unclear. Additionally, increasing number of studies have revealed the role of the tumor microenvironment (TME) in tumor formation and progression [16]. Tumor-infiltrating immune cells (TIC) in the TME contribute directly or indirectly to carcinogenesis and response to chemotherapy or immunotherapy [17,18]. Unfortunately, few studies have been performed on the relationship between RBPs and immune cell infiltration in the TME.

Therefore, we aimed to analyze the differentially expressed RBPs (DERBPs), including AHNAK, MAP1B (microtubule-associated protein 1B), LAMA2 (laminin subunit alpha-2), P4HB (prolyl 4-hydroxylase subunit beta), FASN (fatty acid synthase), and GSDMB (gasdermin B) in BC and explore their prognostic values. A survival-related six RBPs-based prognostic model was established and validated. The close relationship between RBPs and immune cell infiltration was identified. Functionally, genetic knockdown of AHNAK significantly inhibited the proliferation, invasion, and migration of both BC T24 and SW780 cells in vitro and inhibited the formation and growth of subcutaneous tumors in vivo.

## 2. Materials and Methods

### 2.1. Datasets and Patient Information

For the training cohort, clinical information and RNA sequencing data of patients with BC were acquired from the Cancer Genome Atlas (TCGA) [19], which included detailed clinical information regarding OS and complete RNA sequencing data. In total, 414 patients with BC (414 BC tissues and 19 normal bladder tissues) were reflected in the present study. For the validation cohort, the mRNA expression profiles and clinical information for the GSE32548 dataset (146 patients with BC) and GSE31684 dataset (93 patients with BC) were obtained from the Gene Expression Omnibus (GEO) database [20]. Detailed information on patients with BC from the TCGA and GEO databases is listed in Table 1. In addition, data on RBPs were gathered from previously published literature [21]. The present study was approved by the Medical Ethics Committee of Xi’an Jiaotong University (Med-Eth-Re [2021] G-140). No permission was required to use any repository data involved in the present study.

### 2.2. Construction of an Individualized Prognostic Index Based on RBPs

The Linear Models for Microarray Data (LIMMA) package was used to identify the DERBPs in BC, with cut-off values of log2 fold change (FC) greater than 1 and *p* < 0.05. Univariate Cox regression analyses were conducted to select the DERBPs significantly associated with OS in patients with BC. Genes with *p* < 0.05 were utilized for the most minor absolute shrinkage and selection operator. Subsequently, these survival-related genes were tested using multivariate Cox regression analyses. Following multivariate analyses, we established an RBP-based signature to predict the survival of patients with BC. The risk score for each patient was calculated as follows:Risk Score = (β1 × expression of RBP1) + (β2 × expression of RBP2) + (β3 × expression of RBP3) +…+ (βn × expression of RBPn)(1)

All patients from the TCGA and GEO datasets were assigned to the high- or low-risk group based on the median risk score. The Kaplan–Meier technique was used to generate a survival curve. The log-rank test was employed to assess differences in survival rates between the low- and high-risk groups.

### 2.3. Gene Set Enrichment Analysis (GSEA)

To identify the biological processes and pathways correlated with the high- and low-risk groups from the TCGA cohort, GSEA was performed using the clusterProfiler package. Kyoto Encyclopedia of Genes and Genomes (KEGG) [22] and Gene Ontology (GO) gene sets were downloaded from the Molecular Signatures Database [23]. Gene set permutations were run 1000 times for each study to provide a normalized enrichment score (NES), which was used to select pathways enriched in each phenotype. Finally, at *p* < 0.05, the gene set was determined for further research.

### 2.4. Establishment and Assessment of the Nomogram

Univariate and multivariate Cox regression analyses were performed to explore the feasibility of utilizing RBPs-based risk scores as independent predictors of OS in patients with BC in the TCGA cohort. Covariates included age, sex, TNM stage, smoking history, risk score, histological grades, and pathological stages, which were recorded as continuous variables. The nomogram combining the RBPs-based risk score with clinicopathological characteristics, which was correlated with OS, was plotted to predict the 1- and 3-year OS of patients with BC using the ‘rms’ package. The performance of the nomogram was assessed by generating calibration plots. The protein expression of six RBPs was detected using the Human Protein Atlas online database [24]. The prognostic value of the six RBPs in BC was verified using the Kaplan–Meier plotter online tool [25]. Furthermore, the immune microenvironment between the two risk groups of the TCGA cohort was compared. The immune microenvironment was assessed using TIMER2.0 [26], a public database, and seven immune infiltration lymphocyte estimations were obtained.

### 2.5. Cell Culture and Reagents

Human bladder normal epithelial cells (SV-HUC-1) and human BC cell lines (T24 and SW780) were purchased from ATCC (ATCC, Manassas, VA, USA). The T24 cells were grown in McCoy’s 5A (Modified) Medium (Procell Life Science & Technology Co., Ltd., Wuhan, China), and SW780 cells were cultured with Dulbecco’s Modified Eagle Medium (DMEM; Gibco Company, Grand Island, NE, USA) with 10% BI fetal bovine serum (FBS; Biological Industries, Israel) and 1% penicillin/streptomycin (NCM Biotech, Suzhou, China) in 37 °C with 5% CO_2_.

#### 2.5.1. Plasmid Transfection

The target plasmids with Lentivirus packaging were as follows: sh-NC, shAHNAK-A1(TRCN0000123211:5′-CCGGGACCAGAACAAACAGA-AGGAACTCGAGTTCCTTC-TGTTTGTTCTGGTCTTTTTG-3′, shAHNAK-A2(TRCN00-0012321:5′-CCGGCAGCTCT GAAGTGGTTCTGACTCGAGTCAGAACCACTTCAGAGCTGCTTTTTG-3′), shAHNA K-A3(TRCN0000123213:5′-CCGGCCCGTGAAGTCT-TCAGCTCCTCTCGAGAGAGCT GAAGACTTCAC-GGGTTTTTG-3′) (BIOKEEPER Co., Ltd., Xi’an, China). The packaging plasmid psAX2 and envelope plasmid PMD2G were transfected into 293T cells using a PEI-Transfer infection kit (Invitrogen, Carlsbad, CA, USA) following the manufacturer’s instructions. After 48 h, the virus was directly introduced into the cells in a 6-well plate for 24 h. Subsequently, the cell protein was collected after 48 h to measure the infection efficiency.

#### 2.5.2. Cell Viability and 5-Ethynyl-20-Deoxyuridine (EdU) Assays

A CCK-8 assay (TargetMol, Boston, MA, USA) was conducted to measure the cell viability and growth according to the manufacturer’s instructions. Briefly, cells were seeded in 96-well plates at 5 × 10^3^ cells per well. Then, 20 μL of CCK-8 (5 mg/mL) was added to 180 μL of complete medium and incubated for 4 h. The OD reading at a wavelength of 450 nm was measured using an ELISA reader (Bio-Tek, Santa Clara, CA, USA) after shaking for 10 min. For the EdU assay, cells were seeded into confocal plates with a density of 10^5^ cells per well. Subsequently, the cells were incubated with an EdU assay kit (Beyotime Biotechnology, Shanghai, China) at 37 °C for 2 h, fixed with 4% formaldehyde, and permeabilized with 0.1% Triton X-100 following staining of nuclei with Hoechst33342. The results were visualized using a fluorescence microscope. The experiments were performed in triplicate.

#### 2.5.3. Transwell Migration and Invasion Assay

A Transwell chamber (Millipore, Darmstadt, Germany) was used to study cell migration and invasion. Matrigel (Corning, New York, NY, USA) was mixed with serum-free medium at a 1:9 dilution. The cells were resuspended at a density of 10^6^/mL after transfection. The resuspension was then infused into the top chamber at 200 μL. The cells in each chamber were counted under an inverted fluorescent microscope (200× magnification) after 24 h incubation. The cells attached to the top membrane were removed, fixed, stained, and counted again. The experiment was performed in triplicate.

#### 2.5.4. Wound-Healing Assay

AHNAK-knockdown cell lines were tested for migration after they attained a density of 100% in a wound-healing experiment. The distance was marked using a 200 μL pipette tip. Every 24 h until the incision had fully closed, a microscope (200× magnification) was used to take images. The experiment was performed in triplicate to ensure consistent results. To gauge the cells’ potential to migrate, ImageJ software was used to estimate the wound-to-wound distance.

#### 2.5.5. RNA Extraction and Reverse Transcription-Quantitative Polymerase Chain Reaction (RT-qPCR)

Total RNA was extracted from frozen tissues, plasma, and cell cultures using TRIzol reagent (Invitrogen, Carlsbad, CA, USA). RT-qPCR was performed using a previously described method [27]. Briefly, RT-qPCR was performed using TB Green assays (Takara Bio Inc., HeianKyo, Japan) on an ABI 7500 real-time PCR system (Bio-Rad, Hercules, CA, USA), with 18s rRNA as the loading control. The primer sequences (Beijing Tsingke Biotech Co., Ltd., Beijing, China) are listed in Table 2.

#### 2.5.6. Western Blotting Assay

Western blotting assay was performed as described previously [27]. Briefly, the blots were stripped and reprobed to verify equal loading with a specific antibody recognizing β-Actin (ABclonal Technology Co., Ltd., Wuhan, China), Vinculin (Abmart Shanghai Co., Ltd., Shanghai, China). Antibodies against AHNAK, MAP1B, and LAMA2 were purchased from Santa Cruz Biotechnology (Santa Cruz, CA, USA), and antibodies against P4HB, FASN, and GSDMB were obtained from Abcam (Cambridge, UK).

#### 2.5.7. In Vivo Experiments

Six-week-old athymic male BALB/c nude mice were acquired from Beijing Vital River Laboratory Animal Technology Co., Ltd. (Beijing, China). Briefly, mice were divided into four groups (sh-NC, sh-A1, sh-A2, and sh-A3; *n* = 6). T24 cells that had been successfully transfected were resuspended in serum-free medium at a concentration of 5 × 10^6^ cells/0.2 mL and subcutaneously injected in a single side of the posterior flank of mice. Every 2 d, tumor growth and body weight were measured as follows:volume = 0.5 × length × width^2^(2)

The use of mice was permitted by the ethics council of Xi’an Jiaotong University (Permission Number: 2021-674).

#### 2.5.8. Histology and Immunohistochemistry (IHC)

Tumor slices from nude mice xenografts were fixed in 4% paraformaldehyde. In IHC, the primary antibodies were anti-Ki67 (CST, 1:100). Two experienced pathologists evaluated the results based on the intensity of staining (0, 1+, 2+, and 3+) and the proportion of positive cells (0 (0%), 1 (1–25%), 2 (26–50%), 3 (51–75%), and 4 (76–100%)). The staining score was then computed and examined.

### 2.6. Statistical Analysis

Forest plots, receiver operating characteristic (ROC) curves, multiple GSEAs, heat maps, and calibration plots were generated using R Studio (version 3.5.2). A *p* < 0.05 was considered statistically significant.

## 3. Results

### 3.1. Identification of DERBPs

In total, 1617 RBPs were evaluated, and 167 DERBPs met the cut-off (*p* < 0.05, |log2FC| > 1.0). A total of 54 and 113 RBPs were downregulated and upregulated, respectively. The distribution patterns of the RBPs were shown in the heat map in Appendix A.

### 3.2. Identification of DERBPs Associated with Survival in Patients with BC

Thirty-seven OS-associated genes among the DERBPs were screened using univariate Cox regression (*p* < 0.05). Next, the least absolute shrinkage and selection operator (LASSO) regression method was utilized to screen for prognostic RBPs, revealing six RBPs (Figure 1a,b). Finally, the associations between the expression profiles of the six RBPs and the survival of patients with BC were examined using multivariate Cox regression. Six RBPs (AHNAK, MAP1B, LAMA2, P4HB, FASN, and GSDMB) were identified to construct a prognostic signature. These RBPs were classified as risk RBPs (AHNAK, MAP1B, LAMA2, P4HB, and FASN) with hazard ratios (HRs) > 1 or protective RBP (GSDMB) with HRs < 1 (Figure 2e; Table 3).

The survival risk of each patient was assessed by developing a prognostic model as follows:
Risk score = (0.0055 × expression of AHNAK) + (0.0280 × expression of MAP1B) + (0.0536 × expression of LAMA2) + (0.0022 × expression of P4HB) + (0.0037 × expression of FASN) + (0.0508 × expression of GSDMB)(3)

Using the six RBPs signatures, we classified patients with BC in the training cohort into the high- or low-risk group according to the median value (Figure 1c). Figure 1e,g depicted the distribution of risk scores and survival status for each patient.

### 3.3. Application of the Six-RBPs Signature in the Validation Cohort

Kaplan–Meier survival curves revealed that patients with low-risk scores had longer OS than patients with high-risk scores (log-rank test, *p* < 0.0001, Figure 2a). Furthermore, time-dependent ROC curves showed that the AUC values for 1-, 3-, and 5-year OS of the training cohort were 0.72, 0.715, and 0.724, respectively (Figure 2c). The predictive value of the six RBPs signature was further tested using the GSE32548 dataset from the GEO database as an external validation cohort. Patients in the validation cohort were classified into high- or low-risk groups based on the median value (Figure 1d). The risk score and survival distributions for each patient were presented in Figure 1f,h, respectively. Similarly, patients with high-risk scores had a shorter OS than those with low-risk scores in the GSE32548 dataset (*p* = 0.0073; Figure 2b). ROC curves in the time-dependent analysis showed that AUC values for 1-, 3-, and 5-year OS of the variation cohort were 0.786, 0.743, and 0.71, respectively (Figure 2d). Another GSE dataset (GSE31684) with more muscle-invasive BC patients was applied to confirm our conclusion. In this dataset, 27 cases were non-muscle-invasive, and 66 cases were muscle-invasive. As shown in Appendix A, patients with high-risk scores had a shorter OS than those with low-risk scores in the GSE31684 dataset (*p* = 0.0044, hazard ratio = 2.03, 95% CI: 1.12–3.69). ROC curves in the time-dependent analysis showed that AUCs for 3- and 5-year OS were 0.658 and 0.624, respectively (Appendix A). These findings are consistent with our prior validation results, although further studies involving only muscle-invasive BC cases as the test cohort are still needed.

### 3.4. Independent Prognostic Indicator

The predictive ability of the RBP-related risk score for OS in the TCGA cohort was tested by investigating the prognostic value of the six RBPs signatures with several clinicopathological features, including age, sex, TNM (tumor–node–metastasis) stage, smoking history, pathological stage, and histological grade, via univariate and multivariate analyses. Univariate analysis results indicated that age (HR: 1.032, 95% confidence interval (CI): 1.006−1.058, *p* < 0.014), T stage (HR: 1.446, 95% CI: 1.047−1.998, *p* < 0.025), N stage (HR: 1.638, 95% CI: 1.276−2.103, *p* < 0.001), M stage (HR: 2.22, 95% CI: 1.282−6.980, *p* = 0.011), stage (HR: 1.915, 95% CI: 1.381−2.656, *p* < 0.001), and risk score (HR: 1.045, 95% CI: 1.022−1.069, *p* < 0.001) were related to OS (Figure 3a). The risk score was regarded as an independent prognostic indicator in the subsequent multivariate Cox analysis (HR: 1.042, 95% CI: 1.016−1.068, *p* < 0.001; Figure 3c).

### 3.5. Establishment of a Nomogram

A nomogram combining the clinicopathological characteristics associated with OS (age, TNM stage, and stage) and risk score (Figure 3e) was constructed to provide a clinically practical tool to predict the probability of OS in patients with BC. Calibration plots for 1- and 3-year OS were good predictors compared with the ideal model (Figure 3b,d).

### 3.6. GSEA

In the high-risk group, calcium signaling pathway, mitogen-activated protein kinase signaling pathway, focal adhesion, Rap1 signaling pathway, systemic lupus erythematosus, Ras signaling pathway, phosphatidylinositol 3-kinase/AKT signaling pathway, regulation of actin cytoskeleton, transcriptional dysregulation in cancer, and neuroactive ligand-receptor interactions were highly enriched in KEGG pathway analysis. Biological processes, such as microfibril formation, mRNA trans-splicing, the spliced leader (SL) addition, mRNA trans-splicing via the spliceosome, negative regulation of megakaryocyte differentiation, nucleoside-triphosphate diphosphatase activity, protein heterotrimerization, sequence-specific mRNA binding, and translation activator activity, were highly enriched in the high-risk group (Figure 4a,b). We further analyzed the differentially expressed genes between the high- and low-risk groups. A total of 755 differentially expressed genes were observed between two groups (*p* < 0.001, |log2FC| > 1.0, Appendix A).

### 3.7. Verification of the Expression Levels and Prognostic Significance of the Six RBPs

We also tested the mRNA expression level of the six RBPs in BC and paracarcinoma tissues. We found that AHNAK, MAP1B, P4HB, and FASN were significantly upregulated in BC tissues compared with those in paracarcinoma tissues. In contrast, GSDMB was downregulated in BC tissues relative to paracarcinoma tissues. LAMA2 expression did not differ between BC and paracarcinoma tissues (Figure 5a). IHC results from the Human Protein Atlas database were used to verify these observed RBP expressions in BC. As shown in Appendix A, the development of protein expression levels of the six RBPs in BC samples was consistent with the results presented in Figure 5a. Ultimately, protein expressions of the six RBPs were detected in bladder normal epithelial cells (SV-HUC-1) and BC cell lines (T24, SW780). AHNAK, MAP1B, P4HB, and FASN protein expressions were lower in SV-HUC-1 than in T24 and SW780 cells. Contrarily, the expression of GSDMB was higher in SV-HUC-1 than in T24 and SW780 cells, and no significant difference in the expression of LAMA2 was observed in three cells (Figure 3f).

### 3.8. Assessment of Immune Cell Infiltration

The cancer-specific immune microenvironment plays a crucial role in its genesis, progression, and treatment. The online tool TIMER2.0 [26] was used to determine whether there was any difference in the relative abundance of tumor-infiltrating lymphocytes (TILs) between the high- and low-risk groups (Figure 5b). A significant difference was observed between training and control groups in the contents of plasma cells, CD8+ T cells, follicular helper T cells, naive CD4+ T cells, M0 and M2 macrophages, and neutrophils (*p* < 0.05). The proportions of naïve CD4+ T cells, M0 and M2 macrophages, and neutrophils in the high-risk group were higher than those in the low-risk group, whereas the proportions of the plasma cells, CD8+ T cells, follicular helper T cells in the high-risk group were lower than those in the low-risk group. Highly expressed AHNAK in the TME had a negative relationship with tumor purity. AHANK was positively associated with M2 macrophage and neutrophils (Figure 5c).

### 3.9. AHNAK Promotes Cell Proliferation, Migration, and Invasion of BC Cell Lines In Vitro and In Vivo

Among the six RBPs, AHNAK is the most expressed differentially; however, its role in BC has not yet been examined. To explore the biological function of AHNAK in BC, we first characterized the endogenous expression of AHNAK in BC cells based on the predictions in the GEMICCL database [28]. T24 and SW780 cells were observed to have the most abundant AHNAK transcripts and protein expression. Using short hairpin RNA (shRNA), we knocked down AHNAK in T24 and SW780 cell lines (Figure 6a,b). A significant reduction in proliferation was observed in AHNAK knockdown cells (Figure 6c,h,i). Consequently, the results of CCK-8 and EdU experiments suggested that there was a relationship between AHNAK expression and tumor cell proliferation. Furthermore, the effects of AHNAK on cell migration and invasion were examined (Figure 6d–g and Appendix A). AHNAK knockdown inhibited invasion and metastasis in T24 and SW780 cells. In vivo, the tumor volume and weight of AHNAK-knockdown tumors were significantly lower than those of the control group. The proliferation of the tumor was significantly inhibited by AHNAK knockdown (Figure 6j–m and Appendix A). IHC staining showed that Ki67 expression in the knockdown groups was significantly lower than in the control group.

## 4. Discussion

RBPs have important biological functions that directly affect the nucleus, cytoplasm, organelles, proteome complexity, transcriptome, and RNA substrates through various mechanisms [29,30]. Recently, attention has focused on the role of RBPs in the post-transcriptional regulation of gene expression [31,32]. Tristetraprolin belongs to a prognostic family of RBPs that is directly related to post-transcriptional regulation of oncogene mRNAs and exhibits tumor-suppressor properties [33,34]. However, research on the molecular mechanisms of RBP gene interactions in BC has been limited. Multiple studies have suggested that BC is a highly immunogenic malignant tumor and that the TME is involved in BC progression and prognosis [35,36,37]. PD-L1, TMB, and microsatellite instability biomarkers were used to predict the efficacy of treatment [38]. Therefore, a powerful RBP gene signature should be explored and validated for the prediction of BC progression.

In our study, we selected an array of six genes, namely AHNAK, MAP1B, P4HB, FASN, LAMA2, and GSDMB, to create a survival risk score model that can be exploited to predict survival. Using the six RBPs based signature and clinicopathological risk factors in a nomogram, the model was shown to excel at predicting patient survival. Both the AUC and calibration curves showed that our model had good predictability. Combining the risk score with the nomogram improved the ability to segment BC into different risk groups, and facilitating its clinical use. However, both sequencing data for muscle-invasive and non-muscle-invasive BC are included in the GEO dataset. Our conclusions were further validated in another GSE dataset (GSE31684) with more muscle-invasive BC patients. According to the pathway and functional enrichment analyses, the six DERBPs identified in this study may play crucial roles in mRNA trans-splicing, SL addition, mRNA trans-splicing via the spliceosome, negative regulation of megakaryocyte differentiation, nucleoside triphosphate diphosphatase activity, protein heterodimerization, sequence-specific mRNA binding, and translation activator activity.

Among those six RBPs, MAP1B has potential applications in evaluating comprehensive neuropathic paracancer autoantibodies, as well as chemotherapy resistance and cancer progression [39,40]. The establishment of the prognostic model, validation, and detection results in BC tissue all suggest that MAP1B is extremely valuable as a prognostic predictor of BC. Ma et al. and Wu et al. [41,42] claimed that P4HB could be a useful biomarker of cancer. The results of our study are consistent with those of the previous study, and the verification results in BC tissues also suggest that P4HB is highly expressed in BC and has a specific role in the evaluation of prognosis. Tao et al. [43] first discovered the PKM2/FASN axis and hypothesized that the interaction between PKM2 and FASN may be a new target for treatment of BC. In our study, FASN expression in BC was negatively correlated with prognosis and can be used as an independent predictor. The influence of MAP1B and P4HB in tumor immune infiltration cannot be ignored. GSDMB is located in a genomic region that is frequently amplified during cancer development. Therefore, GSDMB function may affect tumor progression and metastasis [44,45]. In our study, GSDMB was downregulated in BC tissues, as shown by the bioinformatics, qPCR, and Western blotting results. The results of our study were different from those previously reported, and this may be related to the heterogeneity of BC. The LAMA2 expression pattern may be helpful as a biomarker to predict prognosis in patients with pituitary neuroendocrine tumors [46]. High expression levels of LAMA2 in the microenvironment are observed to be inversely correlated with tumor purity and positively related to immune cells. Thus, the role of LAMA2 in tumor immune cell infiltration urgently needs to be verified.

As a nucleoprotein, AHNAK is widely expressed in many types of cells. Previous studies suggested that regulating AHNAK expression could affect epithelial-mesenchymal transitions (EMT) in gastric cancer by regulating the Wnt signaling pathway [47]. According to our prognostic model, AHNAK is an independent prognostic factor. Moreover, AHNAK was highly expressed in the BC samples of patients and BC cell lines. High levels of AHNAK indicate poor survival. Interestingly, AHNAK knockdown suppressed the proliferation, migration, and invasion of BC cells. Furthermore, AHNAK knockdown also significantly inhibited the development and growth of tumors in vivo. The obtained data suggest that AHNAK could be used as a prognostic marker of and therapeutic target for BC.

In correlation analysis of AHNAK expression, immune cell infiltration in BC was detected. We believe that the purity of the tumor and multiple immune cells that infiltrate tumors (DC cells, B cells, CD4+ T cells, and CD8+ T cells) were associated with AHNAK expression in BC. We hypothesize that elevated AHNAK levels in the microenvironment would have unfavorable associations with tumor purity. In addition, we established a positive correlation between the level of AHNAK expression and the number of macrophages and neutrophils present in each sample. These outcomes indicate a potential association between AHNAK and immune infiltration of BC. Whether AHNAK interacts with immune cells and influences their functions requires further exploration.

The effectiveness of prediction models for BC of previous studies, such as the nine RBPs constructed by Okholm et al. [13], were not further verified using an external validation set. The six RBPs developed by Wu et al. [14] were previously examined. However, the exact functions of RBPs in BC have not been further explored. In our study, a close relationship between the six RBPs and tumor immune infiltrating cells were identified, although further studies are needed to illustrate the underlying molecular regulatory mechanisms. Among those six DERBPs, AHNAK, which was identified in both Wu’s model [14] and our prediction model, received considerable attention from us. Interestingly, genetic knockdown of AHNAK significantly inhibited the proliferation, invasion, and migration of both T24 and SW780 cells in vitro and inhibited the formation and growth of subcutaneous tumors in vivo, suggesting the oncogenic role of AHNAK in BC. Further studies, particularly mechanistic studies, are needed to determine the specific role of other RBPs.

RBPs have recently been attracting increased attention for the treatment of cancers. Accumulating evidence suggests the potential effects of therapeutic drugs on RBP functions. Pyrvinium pamoate, an FDA-approved anthelminthic drug, was reported to promote nuclear import of HuR (an RBP), leading to unbearable genomic instability and cell death in BC cells [48]. Similarly, the RBP EWSR1 was identified to aggregate in the cytoplasm in temozolomide (a chemotherapeutic agent)-resistant glioblastoma cells and patient samples [49]. Additionally, the RBP MEX3B was proven to mediate resistance to cancer immunotherapy by down-regulating HLA-A expression on the surface of tumor cells [50]. Interestingly, our study also demonstrated that highly expressed AHNAK in the tumor microenvironment has a negative relationship with tumor purity, suggesting a potential link between RBPs and immune cell infiltration. Considering the vital role of immunotherapy and chemotherapy in BC treatment, further in-depth studies are needed to uncover the exact relationship between immunotherapy/chemotherapy and RBP in BC.

## 5. Conclusions

In this study, we thoroughly analyzed the key RBPs associated with OS in patients with BC using bioinformatic analysis. Moreover, we identified six DERBPs in BC. Survival analysis revealed that the six RBPs may have a predictive value for BC. A close relationship between RBPs and immune cell infiltration was identified. More importantly, we functionally confirmed the oncogenic role of AHNAK in BC (Figure 7). Further research is needed to explore the roles of these six RBPS in BC cells and their potential applications as prognostic biomarkers in BC.

## Figures and Tables

**Figure 1 cancers-14-03739-f001:**
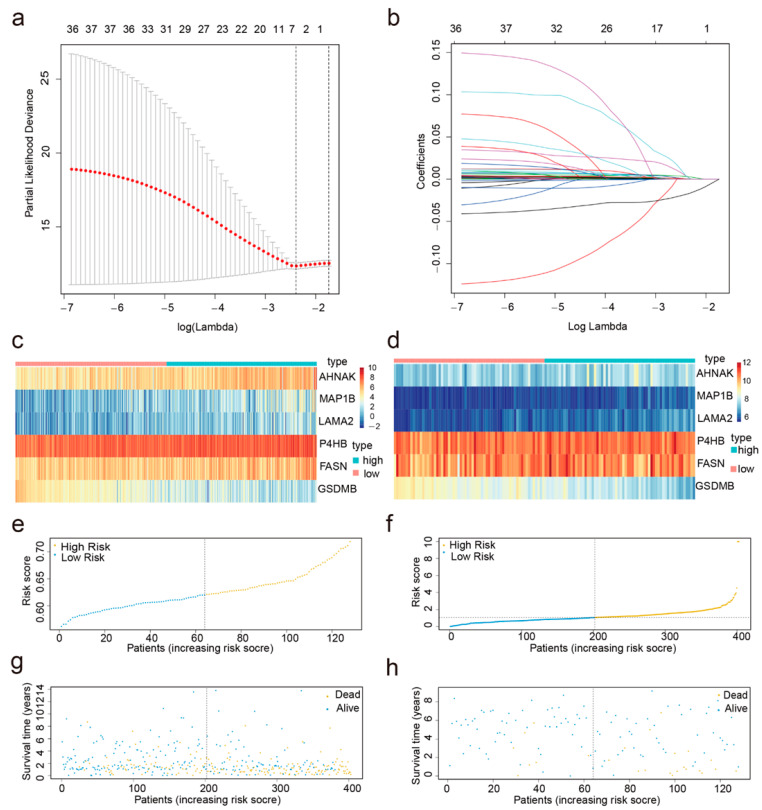
Predictor selection using the most minor absolute shrinkage and selection operator (LASSO). (**a**) Parameter (lambda) selection by the LASSO model adopted 10-fold cross-validation via minimum criteria; (**b**) LASSO coefficient profile plot of the six RBPs gene pairs against the log (lambda) sequence risk curve of training and test sets; (**c**) heat map of the six RBPs for the high- and low-risk groups in the training cohort; (**d**) heat map of the six RBPs for the high- and low-risk groups in the validation cohort; (**e**) risk score distribution for the training cohort; (**f**) risk score distribution for the validation cohort; (**g**) distribution of the survival status of the training cohort; (**h**) survival status distribution of the variation cohort.

**Figure 2 cancers-14-03739-f002:**
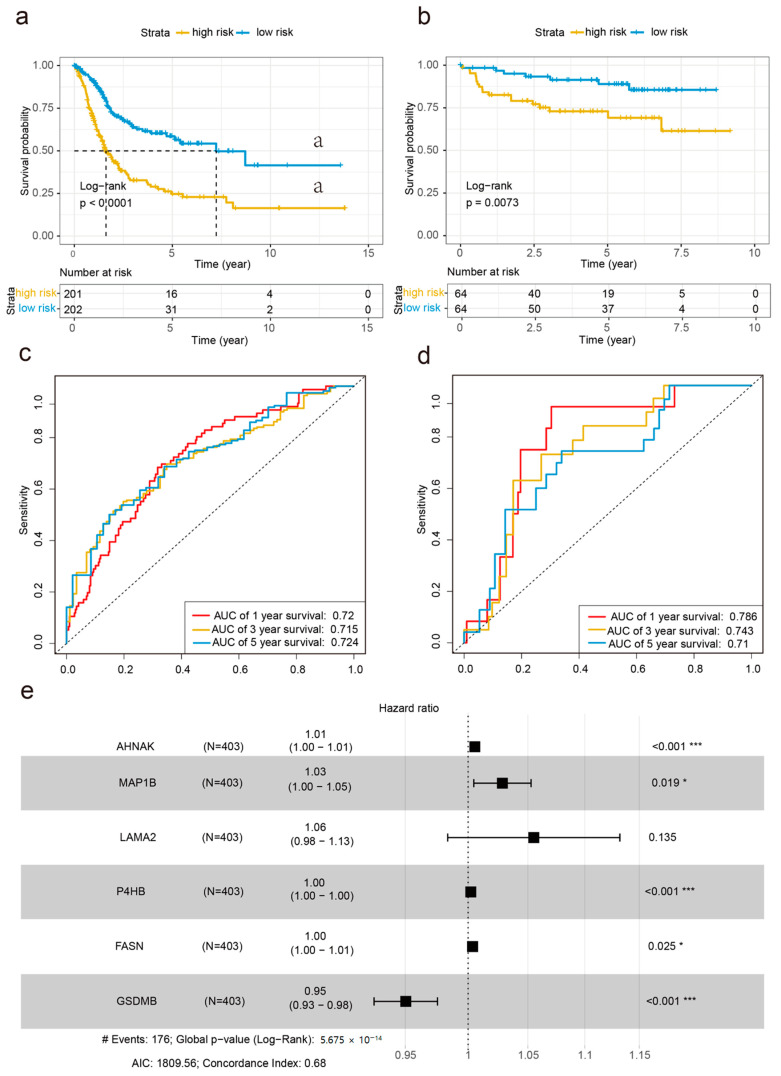
DEGs (differentially expressed genes) of RBPs were utilized to build prognostic models and analyze and verify GEO datasets. (**a**) Survival analysis of the training cohort. (**b**) The survival analysis curve of the six RBPs in the validation cohort derived by creating the prognostic model shown in the forest map; yellow indicates the patients in the high-risk group, and blue indicates the patients in the low-risk group. (**c**) Receiver operating characteristic (ROC) curve for the training cohort. (**d**) ROC curve for the validation cohort. (**e**) The six RBPs were derived using the prognostic model depicted in the forest map. * *p* < 0.05, *** *p* < 0.001.

**Figure 3 cancers-14-03739-f003:**
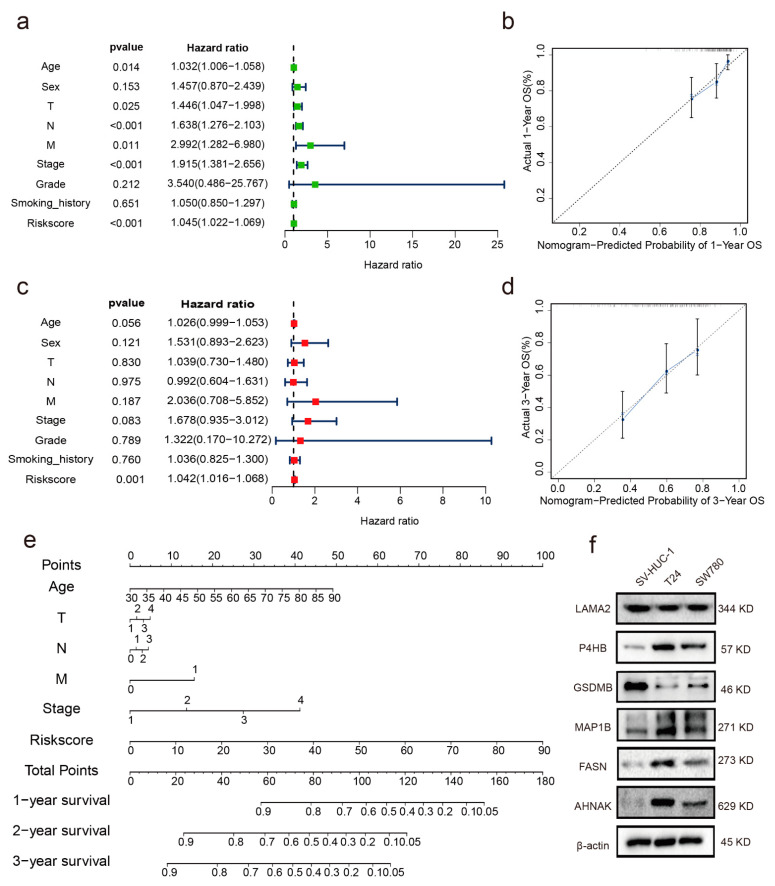
Independent prognostic analysis and prediction of 1- and 3-year nomograms for patients with BC in the training and validation cohorts. (**a**) Single-factor prognostic analysis for the training cohort. (**b**) Nomogram for the prediction of 1-year survival probability of patients with BC in the training set. (**c**) Multifactor prognosis analysis for the training cohort. (**d**,**e**) Nomograms for the prediction of 2- and 3-year survival probabilities for patients with BC in the training cohort. (**f**) Verification of protein expression levels of the six RBPs.

**Figure 4 cancers-14-03739-f004:**
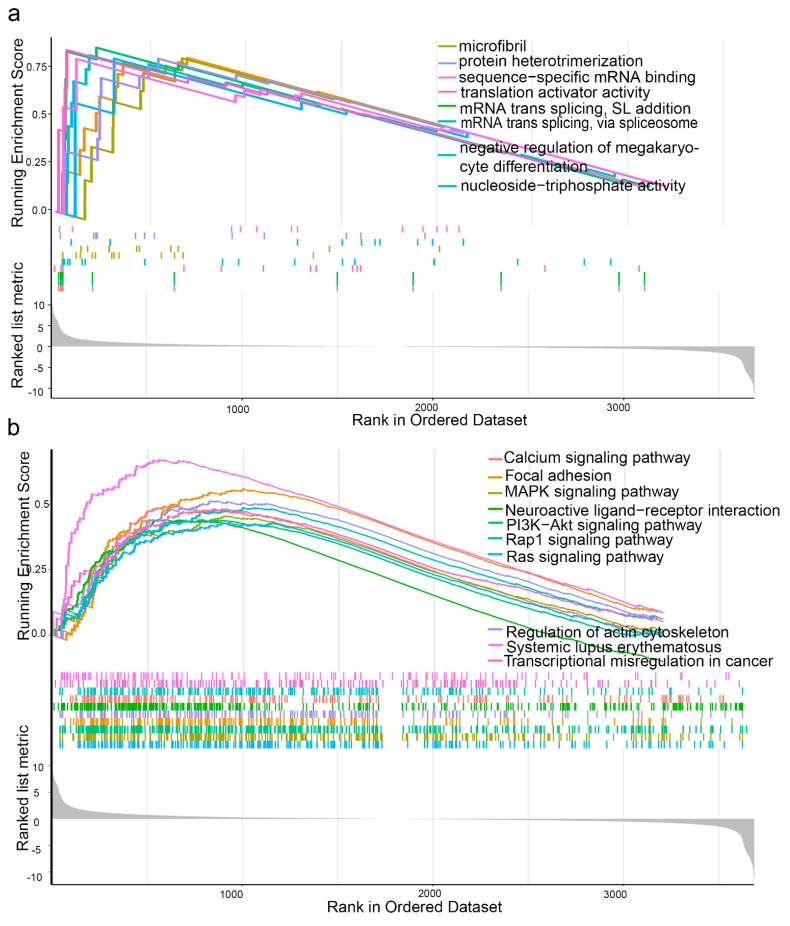
(**a**) Gene ontology gene set enrichment analysis; (**b**) Kyoto Encyclopedia of Genes and Genomes gene set enrichment analysis.

**Figure 5 cancers-14-03739-f005:**
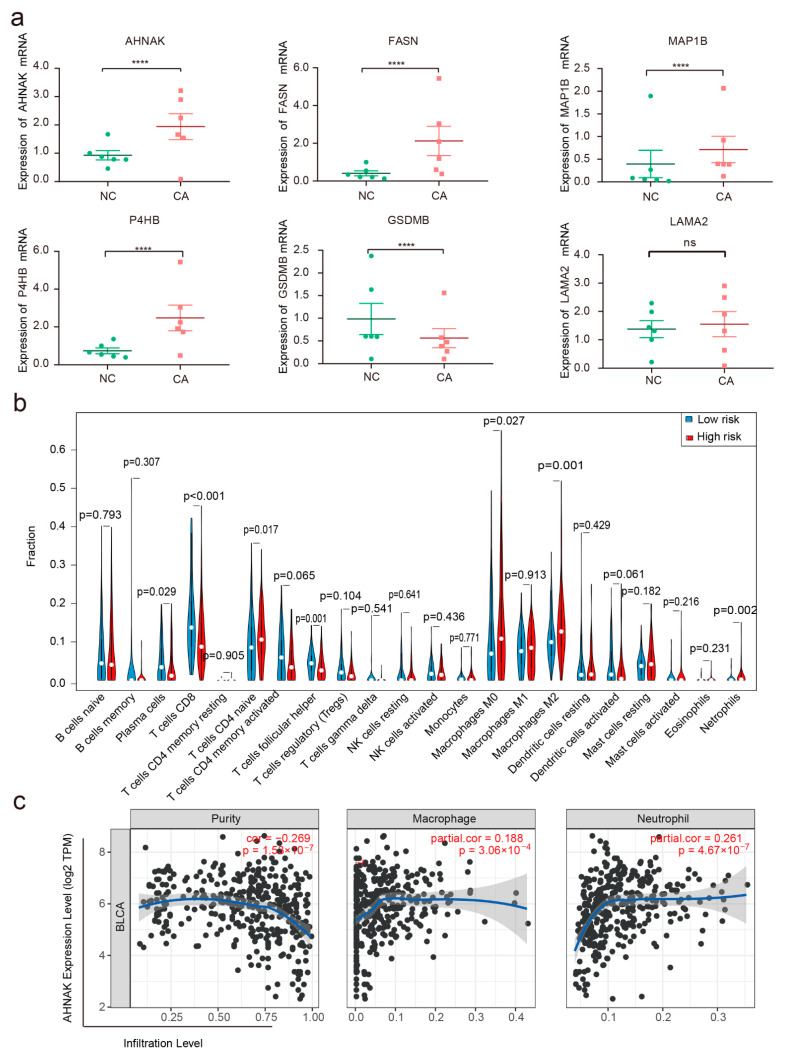
Expression of the six RBPs in tumor tissue and the immune microenvironment. (**a**) The RNA expression patterns of the six RBPs in BC types and paired non-tumor samples. Each red dot represents a distinct tumor sample, and each green dot represents a non-tumor sample. (**b**) Differential analysis of tumor-infiltrating immune cells between the high- and low-risk groups; red: high-risk group, blue: low-risk group (*p* < 0.05). (**c**) Visualization of the correlation of AHNAK expression with macrophage and neutrophil immune infiltration levels in BC (*p* < 0.05). **** *p* < 0.0001.

**Figure 6 cancers-14-03739-f006:**
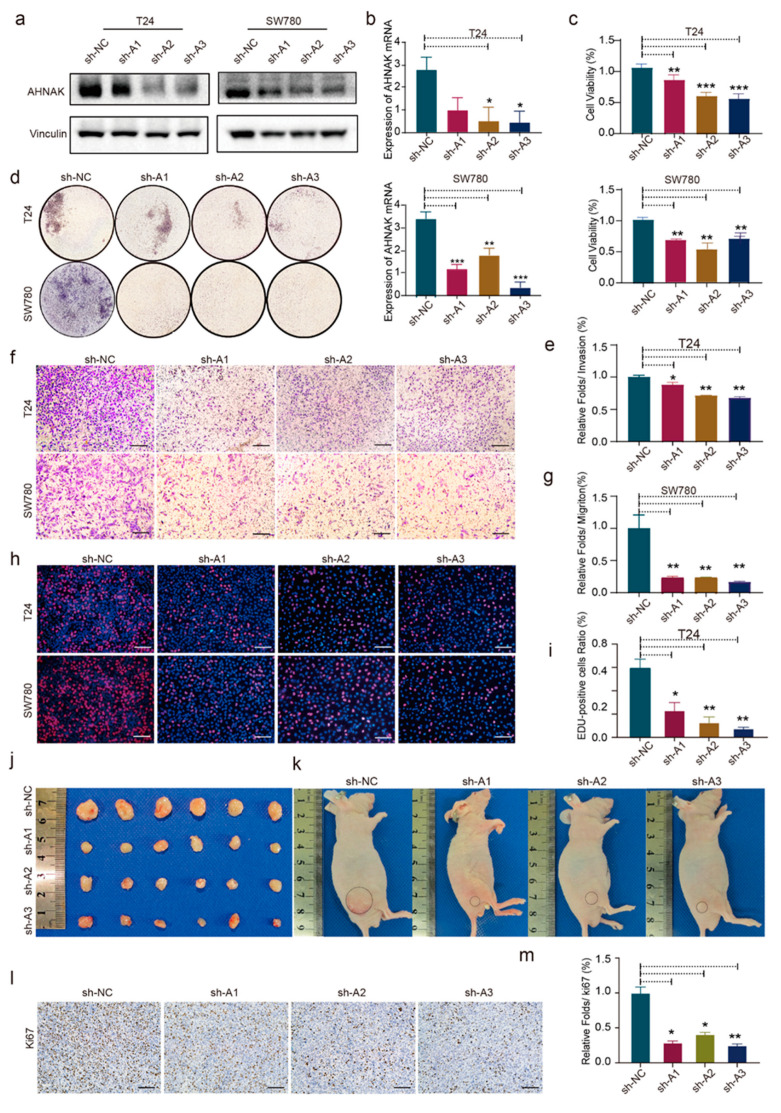
AHNAK accelerates BC cell proliferation, migration, and invasion in vitro and in vivo. (**a**,**b**) Western blotting and PCR results of the AHNAK-knockdown T24/SW780 cells. (**c**) Cell proliferation of AHNAK-knockdown T24/SW780 cells was determined using the CCK-8 assay (mean ± SD, *n* = 6); Transwell assays showed that AHNAK knockdown significantly inhibited (**d**,**e**) invasion and (**f**,**g**) migration of T24 and SW780 cells. (**h**,**i**) An EdU assay kit was used to test the cell proliferation ability. (**j**,**k**) Knockdown of AHNAK inhibited the proliferation of T24 cells in vivo. (**l**,**m**) Immunohistochemical staining demonstrated the suppression of AHNAK-knockdown cells in vivo, as indicated by the expression of Ki67-positive cells. * *p* < 0.05, ** *p* < 0.01, *** *p* < 0.001; Scale bar: 100 μm.

**Figure 7 cancers-14-03739-f007:**
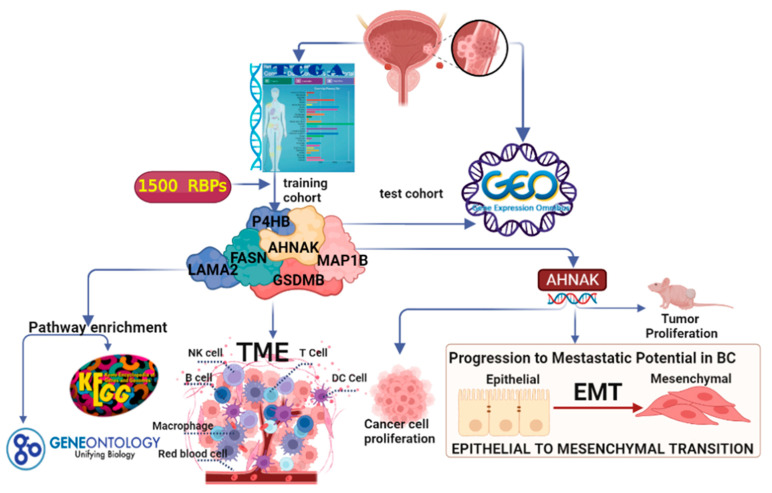
Flow chart of data processing in this study. TCGA, The Cancer Genome Atlas; GEO, Gene Expression Omnibus Database; GO, gene ontology; KEGG, Kyoto Encyclopedia of Genes, and Genomes; TIMER2.0 (Tumor Immune Estimation Resource 2.0).

**Table 1 cancers-14-03739-t001:** Detailed information on patients with BC from the Cancer Genome Atlas (TCGA) and Gene Expression Omnibus (GEO) databases.

Characteristic		TCGA	GEO (GSE32548)	GEO (GSE31684)
Age	<60	88	32	19
	>60	324	114	74
Sex	Female	108	34	25
	Male	304	112	68
Stage	Ta	0	49	8
	T1	11	53	19
	Tis	0	1	55
	Tx	6	1	10
	≥T2	395	42	1
	0	0	1	0
Grade	G1	3	19	0
	G2	23	46	6
	G3	386	80	87
Survival status	dead	112	119	65
	living	217	26	28
	unknown	83	1	0

**Table 2 cancers-14-03739-t002:** Primers used in the present study.

Primer Name	Sequence 5′–3′	
AHNAK	Forward	CTCGTCGCCGCCAGTAG
	Reverse	TCTTTGCAGGATTCCGCTCA
MAP1B	Forward	AATTCCTGGGCAAACTGGTCT
	Reverse	AGAGCCGGACTGGAGAATGA
LAMA2	Forward	GGCTTCCGTTGTCAGCAATC
	Reverse	CAAGTTTCTCAGCGTTGGCA
P4HB	Forward	TCATCGCCAAGATGGACTCG
	Reverse	CCACCGCTCTCCAGGAATTT
FASN	Forward	ACCTCCGTGCAGTTCTTGAG
	Reverse	GTTCAGGATGGTGGCGTACA
GSDMB	Forward	AGACGATGAGAAAGTCTTTGGGT
	Reverse	TAGCTCCCCGGAAATCAGGA

**Table 3 cancers-14-03739-t003:** Six RNA-binding proteins and their hazard ratios.

ID	coef	HR	HR.95L	HR.95H	*p* Value
AHNAK	0.005486	1.005501	1.00313	1.007878	5.26 × 10^−6^
MAP1B	0.028034	1.028431	1.004654	1.05277	0.018821
LAMA2	0.053639	1.055104	0.983493	1.131929	0.134704
P4HB	0.002214	1.002217	1.000971	1.003465	0.000487
FASN	0.003685	1.003692	1.000466	1.006929	0.024866
GSDMB	−0.05084	0.950431	0.926115	0.975385	0.000121

## Data Availability

All publicly available datasets used in this study were described in the Section 2. Other datasets generated in this study are available upon reasonable request to the corresponding author.

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
