# Peer review of "Integrated Analysis and Identification of Critical RNA-Binding Proteins in Bladder Cancer"

_cancers, 2022, doi:10.3390/cancers14153739_

Round 1

Reviewer 1 Report

The authors used publicly available databases to detect RNA-binding proteins (RBPs) associated with overall survival in bladder cancer patients. The risk score calculated using six RBPs further classified bladder cancer into high-risk and low-risk cases, and showed a significant difference in overall survival between them.

The authors first used bladder cancer cases in TCGA as the training cohort and then used GEO cases as the test cohort, noting that even in the test cohort, the classification of cases by risk score was indicative of patient prognosis.

I have serious concerns about this study, because most of the TCGA cases are muscle-invasive bladder cancer, while more than half of the GEO cases are non muscle-invasive bladder cancer. Muscle invasive bladder cancer and non muscle-invasive bladder cancer are basically different in pathophysiology, prognosis, and treatment. To test the hypothesis that risk scores using RBPs are useful in predicting patient prognosis, the test cohort should use only muscle-invasive bladder cancer, for example, among GEO cases. The test cohort should be limited to GEO cases with muscle-invasive bladder cancer.

Author Response

Replies to Reviewer 1:

The authors used publicly available databases to detect RNA-binding proteins (RBPs) associated with overall survival in bladder cancer patients. The risk score calculated using six RBPs further classified bladder cancer into high-risk and low-risk cases and showed a significant difference in overall survival between them. The authors first used bladder cancer cases in TCGA as the training cohort and then used GEO cases as the test cohort, noting that even in the test cohort, the classification of cases by risk score was indicative of patient prognosis.

I have serious concerns about this study, because most of the TCGA cases are muscle-invasive bladder cancer, while more than half of the GEO cases are non-muscle-invasive bladder cancer. Muscle invasive bladder cancer and non-muscle-invasive bladder cancer are basically different in pathophysiology, prognosis, and treatment. To test the hypothesis that risk scores using RBPs are useful in predicting patient prognosis, the test cohort should use only muscle-invasive bladder cancer, for example, among GEO cases. The test cohort should be limited to GEO cases with muscle-invasive bladder cancer.

Response: Thank you for your insightful suggestions. We agreed with the reviewer that most of the TCGA cases are muscle-invasive bladder cancer, while more than half of the GEO cases are non-muscle-invasive bladder cancer. We have carefully analyzed the data downloaded from the TCGA and GEO (GSE32548 dataset) databases. All the cases from TCGA were muscle-invasive bladder cancer in our study. As for the data from GEO GSE32548 dataset, 90 cases were non-muscle-invasive and 36 cases were muscle-invasive.

We also agreed with the reviewer that the test cohort should use only muscle-invasive bladder cancer among GEO cases. But the fact is that it is difficult to obtain the original clinical data and parameters of muscle-invasive bladder cancer from the original authors. So we used another GSE dataset with more muscle-invasive bladder cancer patients to confirm our conclusion. Accordingly, the predictive value of the six-RBPs signature was further tested using the GSE31684 dataset from the GEO as an external validation cohort of muscle-invasive bladder cancer. In this dataset, 27 cases were non-muscle-invasive and 66 cases were muscle-invasive. As shown in figure S3, patients with high-risk scores had shorter OS than those with low-risk scores in the GSE31684 dataset (P=0.0044, Hazard Ratio=2.03, 95% CI:1.12-3.69). ROC curves in the time-dependent analysis showed that AUCs for 3-, and 5-year OS were 0.658 and 0.624, respectively. The findings are consistent with our prior validation results, although further studies involving only muscle-invasive bladder cancer cases as the test cohort are still needed. We have added those new data in our revised manuscript (figure S3).(Please see the attachment)

Reviewer 2 Report

The authors assessed the potential role and function of RNA-binding proteins that may play in bladder cancer based on the reliable datasets from The Cancer Genome Atlas and Gene Expression Omnibus and further demonstrated the function of the identified proteins crucial to the development of bladder cancer by using multiple well-established assays. The data are in great support of the final conclusion in the manuscript that the identified RNA-binding proteins is critical to the proliferation of bladder cancer cells. However, there are a few minor recommendations for the authors before the final acceptance of the manuscript. 

1. The title of the manuscript can be more specific and conclusive. For instance, the authors may consider "Integrated analysis and identification of critical RNA-binding proteins in bladder cancer". 

2. The authors can provide a schematic image for the final conclusions to facilitate the comprehension of the data. 

3. For the patients' data, the readers in the journal may wonder if there was any drug taken by the patients. Is it possible to provide some discussions on the medications and how the medicine use will affect the final conclusions? 

Author Response

Replies to Reviewer 2:

The authors assessed the potential role and function of RNA-binding proteins that may play in bladder cancer based on the reliable datasets from The Cancer Genome Atlas and Gene Expression Omnibus and further demonstrated the function of the identified proteins crucial to the development of bladder cancer by using multiple well-established assays. The data are in great support of the final conclusion in the manuscript that the identified RNA-binding proteins are critical to the proliferation of bladder cancer cells. However, there are a few minor recommendations for the authors before the final acceptance of the manuscript. 

Response: Thank you for your positive comments.

  1. The title of the manuscript can be more specific and conclusive. For instance, the authors may consider "Integrated analysis and identification of critical RNA-binding proteins in bladder cancer". 

Response: Thank you for your suggestion. The title of the manuscript has been changed to "Integrated analysis and identification of critical RNA-binding proteins in bladder cancer" as suggested.

  1. The authors can provide a schematic image for the final conclusions to facilitate the comprehension of the data. 

Response: Thank you for your suggestion. A schematic image for the final conclusions of our study was added to our revised manuscript (Figure 7).

  1. For the patients' data, the readers in the journal may wonder if there was any drug taken by the patients. Is it possible to provide some discussions on the medications and how the medicine use will affect the final conclusions? 

Response: Thank you for raising this critical issue. The raw RNA sequencing data were downloaded from the TCGA and GEO (GSE32548 dataset) databases. We have carefully evaluated the databases, but disappointedly, all the patients in the databases did not receive drug treatment.

We agreed with the reviewer that exposure to some certain drugs might result in the regulation of the expression or function of RNA-binding proteins in bladder cancer, which will consequently affect our final conclusions. RNA binding proteins have been attracting increased attention in the treatment of cancers recently. Indeed, accumulating evidences have suggested the potential effects of therapeutic drugs on RNA-binding proteins functions. It was reported that pyrvinium pamoate, an FDA-approved anthelminthic drug, promoted nuclear import of HuR (an RNA-binding protein), leading to unbearable genomic instability and cell death in bladder cancer cells (Guo J et al. Oncotarget. 2016). Similarly, RNA binding protein EWSR1 was identified to aggregate in the cytoplasm in Temozolomide (a chemotherapeutic agent)-resistant glioblastoma cells and patient samples (Tiek DM et al. Sci Adv. 2022). Additionally, the RNA-binding protein MEX3B was proved to mediate resistance to cancer immunotherapy by downregulating HLA-A expression on the surface of tumor cells, thereby making the tumor cells unable to be recognized and killed by T cells (Huang L et al. Clin Cancer Res. 2018). Interestingly, our study also demonstrated that highly expressed AHNAK in the tumor microenvironment has a negative relationship with tumor purity, suggesting a potential link between RNA-binding protein and immune cell infiltration. Considering the vital role of immunotherapy and chemotherapy in bladder cancer treatment, further in-depth studies are needed to uncover the exact relationship between immunotherapy/chemotherapy and RNA-binding protein in bladder cancer. We have added those related statements in the discussion section of our revised manuscript (Discussion section, para 7).

Round 2

Reviewer 1 Report

This article is acceptable.